# Efficacy and Safety of Liver Chemoembolization Procedures, Combined with FOLFIRI Chemotherapy, in First-Line Treatment of Metastatic Colorectal Cancer in Patients with Oncogene Mutations

**DOI:** 10.3390/cancers16010071

**Published:** 2023-12-22

**Authors:** Marcin Szemitko, Aleksander Falkowski, Monika Modrzejewska, Elzbieta Golubinska-Szemitko

**Affiliations:** 1Department of Interventional Radiology, Pomeranian Medical University, Al. Pow. Wielkopolskich 72, 70-111 Szczecin, Poland; zrz@pum.edu.pl; 2II Department of Ophthalmology, Pomeranian Medical University, Al. Pow. Wielkopolskich 72, 70-111 Szczecin, Poland; monika_modrzej@op.pl; 3Department of General and Dental Diagnostic Imaging, Pomeranian Medical University, Al. Pow. Wielkopolskich 72, 70-111 Szczecin, Poland; e.golubinska@gmail.com

**Keywords:** DEB-TACE, FOLFIRI, FOLFOX, irinotecan, microspheres

## Abstract

**Simple Summary:**

Colorectal cancer (CRC) is the the second leading cause of cancer-related mortality. The aim of the study was to evaluate the efficacy and safety of transarterial chemoembolization (DEB-TACE) procedures for the treatment of metastatic liver lesions from CRC, using irinotecan-loaded microspheres as first-line treatment together with FOLFIRI chemotherapy. The response and disease control rates were 55% and 80%, respectively. The toxicity profiles among our patients were generally acceptable and no serious events were recorded. The combination of FOLFIRI with chemoembolization may provide a valuable treatment option for patients with oncogene mutations who do not qualify for monoclonal antibody therapy.

**Abstract:**

Purpose. The usual first- and second-line treatments for inoperable liver metastases from colorectal cancer (CRC) involve systemic chemotherapy, often with molecular targeted therapy. Chemoembolization, using microspheres loaded with irinotecan, has also been available as a treatment option for many years, used mainly in later lines of treatment when, due to increasing resistance, other chemotherapy regimens may have been exhausted. However, when there are contraindications to molecular therapies, the use of chemoembolization as first or second lines of treatment, in combination with FOLFIRI chemotherapy, may provide greater efficacy due to reduced irinotecan resistance. Objective. The aim of the study was to evaluate the efficacy and safety of transarterial chemoembolization (DEB-TACE) procedures for the treatment of metastatic liver lesions from CRC, using irinotecan-loaded microspheres as first-line treatment together with FOLFIRI chemotherapy. Patients and methods. The analysis included 20 patients (12 females; 8 males) with unresectable liver metastases in the course of CRC with KRAS, NRAS and BRAF mutations, who underwent 73 chemoembolization procedures with microspheres loaded with 100 mg of irinotecan, in combination with interspersed FOLFIRI chemotherapy. Response to treatment was assessed through computed tomography according to the Modified Response Evaluation Criteria in Solid Tumors (mRECIST). Progression-free survival (PFS) and overall survival (OS) were calculated using the Kaplan–Meier method. Assessment of adverse events utilized the Cancer Therapy Evaluation Program’s Common Terminology Criteria for Adverse Events (CTCAE; version 5.0). Results. Partial remission (PR) was observed in 11 (55%) patients while 5 (25%) patients showed stable disease (SD). Progression (PD) was observed in 4 (20%) patients. Median PFS was 9.1 months (95% CI: 7.2–10.1 months) and median OS was 20.7 months (95% CI: 18.2–23.3 months). The most common adverse events (AEs) resulting in treatment delay were hematological disorders, notably neutropenia (CTCAE grades 1–3). No deaths or AEs above grade 3 occurred during TACE. Continued FOLFIRI chemotherapy after TACE treatments resulted in grade 4 neutropenia in two patients, grade 3 in four patients and grade 2 thrombocytopenia in two patients. Conclusion. Combining FOLFIRI chemotherapy with chemoembolization procedures for liver metastatic lesions from colorectal cancer may provide a valuable treatment option for patients not qualified for monoclonal antibody therapy.

## 1. Introduction

Liver chemoembolization with irinotecan-loaded microspheres (i.e., drug-eluting bead transarterial chemoembolization; DEB-TACE) has been available as a treatment option for inoperable liver metastases from colorectal cancer (CRC) for many years [1]. However, the preferred palliative treatments remain as systemic chemotherapy regimens based on folinic acid, 5-fluorouracil and either irinotecan or oxaliplatin. Parallel advances, particularly with the introduction of biologic therapies, have resulted in significant increases in the effectiveness of systemic chemotherapy [2]. Current standard drugs are therefore based on cytotoxic chemotherapy, VEGF and EGFR antibodies and multikinase inhibitors [3]. 

One problem results from the fact that patients with contraindications to the above molecular targeted therapies are usually given a standard systemic chemotherapy regimen, with markedly lower resulting efficacy [4]. Moreover, patients with mutations in the *RAS* or *BRAF* genes have significantly worse outcomes [5,6]. 

The liver is most often a critical organ in terms of patient survival and one option that may increase the effectiveness of FOLFIRI (5-fluorouracil/leucovorin and irinotecan) chemotherapy, especially with metastatic lesions of the liver, is to combine its use with chemoembolization procedures (DEB-TACE). 

Previous studies have shown variability in the efficacy of chemoembolization for metastases from CRC. Most studies have analyzed chemoembolization procedures in later lines of treatment when, due to increasing cell resistance to previously administered chemotherapeutics, their effectiveness may have been reduced [7,8]. 

In the present study, the efficacy of treatment and incidence of side effects were evaluated during and after alternating liver chemoembolization treatments with irinotecan-loaded microspheres as first-line treatment together with FOLFIRI systemic chemotherapy, in patients with *KRAS, NRAS* and *BRAF* oncogene mutations and inoperable liver metastases from colorectal cancer (CRC-LM).

## 2. Materials and Methods

The study included 20 patients (12 females and 8 males) with unresectable liver metastases from *KRAS*-, *NRAS*- and *BRAF*-mutated colorectal cancer, qualified for first-line, palliative, FOLFIRI chemotherapy. All patients were ineligible for simultaneous bevacizumab monoclonal-antibody treatment, mainly due to a high risk of thromboembolic events. 

FOLFIRI chemotherapy and DEB-TACE treatment regimens were administered between November 2015 and December 2020. The study was approved by the Bioethics Committee of the Pomeranian Medical University, Szczecin, Poland. Patients were qualified for FOLFIRI combined with drug-eluting bead transarterial chemoembolization (DEB-TACE) treatments after consultation with an oncologist, who administered the FOLFIRI chemotherapy. 

Treatment was discussed in detail with all patients and informed consent was obtained from them prior to initiation.

Exclusion criteria: liver failure, bilirubin > 3 mg/dL; renal failure, creatinine > 2 mg/dL; thrombocytopenia, platelets < 100 × 10^9^/L; tumor involvement of >50% of liver parenchyma; history of anaphylactic reaction to contrast; Eastern Cooperative Oncology Group (ECOG) performance status > 1; patient’s eligibility for concomitant treatment with monoclonal antibodies.

One week prior to DEB-TACE chemoembolization, patients received FOLFIRI palliative i.v. chemotherapy: 400 mg/m^2^ (body surface area) of 5-fluorouracil (on days 1 and 2 of the cycle); 200 mg/m^2^ of leucovorin (on days 1 and 2); and 180 mg/m^2^ of irinotecan (on day 1).

The first DEB-TACE chemoembolization procedure, on the right lobe of the liver, was performed on day 7 or 8 after the first FOLFIRI treatment. A week after the second FOLFIRI treatment, the next DEB-TACE treatment was performed on the left lobe of the liver. After the 3rd and 4th FOLFIRI treatments, chemoembolization of the right and then left lobes was repeated, respectively. Depending on the patient’s condition, tolerance and laboratory blood parameters, the intervals between FOLFIRI chemotherapy treatments and DEB-TACE procedures were modified in some patients. In the case of single liver-lobe involvement, only two DEB-TACE treatments were performed at four-week intervals, with two FOLFIRI treatments before the second chemoembolizations. After the last DEB-TACE procedure, patients continued with only the FOLFIRI systemic chemotherapy. 

Microspheres (Embozene Tandem 100 µm, 2 mL (CeloNova Biosciences, now Varian Medical System, Palo Alto, CA, USA)) were loaded with 100 mg of irinotecan according to the manufacturer’s instructions and mixed with 5 mL of a contrast agent (Iodixanolum at 320 mg I/mL) immediately before liver administration.

The DEB-TACE procedures were performed by interventional radiologists with certified skills and many years of experience in interventional radiology.

Patients received a prophylactic antibiotic (cefazolin), a proton-pump inhibitor, a steroid and, on the day of chemoembolization, an antiemetic drug and an infusion of 1 L of 0.9% NaCl, according to hospital guidelines.

### 2.1. Chemoembolization Procedure

The Seldinger method was used to puncture the right or left common femoral artery and the celiac trunk was catheterized (in the case of an anatomical variant of the hepatic artery visible from previous computed tomography (CT) scan, the superior mesenteric artery was also catheterized) using a SIM 5F catheter (Cordis, Miami Lakes, FL, USA). Arteriography and cone-beam CT were performed and liver vascularization was assessed. A microcatheter (Progreat^®^ 2.7F, Terumo, Tokyo, Japan) was used for super-selective catheterization of the right or left branch of the hepatic artery. Prior to microsphere injection through the microcatheter, 2 mL of lidocaine was administered. Then, under fluoroscopy, the mixture of the microspheres and contrast agent was administered at a rate of approximately 1 mL/min, avoiding embolizate reflux. The embolizate was administered until “near-stasis” (stasis that resolved within a few seconds) was achieved at the level of the vessels supplying the tumors or until the entire dose of embolizate had been administered.

Pain, during and after the procedure, was controlled with opioid infusion and non-steroidal anti-inflammatory agents. Prophylactic dexamethasone (8 mg i.v.) and an antibiotic (cefazolin, 1 g i.v.) were administered twice daily and, in the case of vomiting, ondansetron (8 mg i.v.).

Most patients were discharged from the hospital the day after the procedure.

Adverse effects and complications occurring perioperatively and within 30 days after surgery were assessed according to the standards and terminology of the Cancer Therapy Evaluation Program’s Common Terminology Criteria for Adverse Events (CTCAE, version 5.0).

Data were recorded (Excel 2007; Microsoft, Washington, DC, USA) in a form suitable for statistical evaluation.

### 2.2. Imaging and Tumor Response

To evaluate the response to treatment, triphasic CT scans were performed 3–4 weeks after the last TACE treatment and were compared with those before treatment. Subsequent CT scans were performed at three-month intervals. Response was assessed using the Modified Response Evaluation Criteria in Solid Tumors (mRECIST), which are the most often used criteria to assess tumor response in CRC-LM after DEB-TACE treatment. This method assesses tumor response using identification of post-embolization intratumoral necrotic areas and the reduction in tumor burden as seen in triphasic CT. Partial response was defined as at least a 30% decrease in the sum of diameters of viable (enhanced in the arterial phase) target lesions—and progressive disease as an increase of at least 20% in the sum of the diameters of viable (enhanced) target lesions, taking as a reference the smallest sum of the diameters of viable (enhanced) target lesions since the start of treatment. Stable disease was defined as any case that did not qualify for either partial response or progressive disease.

### 2.3. Statistical Analyses

Descriptive statistics of the studied variables were given as means or medians and ranges. 

Progression-free survival (PFS) and overall survival (OS) were calculated using the Kaplan–Meier method from the date of the patient’s first FOLFIRI treatment. All statistical tests were performed using commercial software (SPSS, Chicago, IL, USA; version 26). 

## 3. Results

### 3.1. Baseline Characteristics

The patients (*n* = 20) underwent 73 chemoembolization treatments (interspersed with FOLFIRI systemic chemotherapy) between the inclusive dates of November 2015 and December 2020. 

Two patients with only the involvement of the left lobe of the liver had only two TACE procedures each. In 16 patients with involvement of both lobes, four TACE procedures were performed for each patient. In one patient, due to side effects with the involvement of both lobes, two TACE procedures were performed and in another patient three procedures were performed. 

Patient characteristics are presented in Table 1.

The technical success of the TACE procedures was 100%. In 16 procedures with the right liver lobe and 37 procedures with the left lobe, early near-stasis occurred and in order to avoid permanent hepatic-artery occlusion with these procedures, approximately 50% of the embolizate dose was administered. In the remaining treatments, the entire dose of the embolizate was successfully administered. Following cycles, which included TACE, patients were given FOLFIRI systemic chemotherapy cycles without TACE and the mean number of FOLFIRI cycles per patient was 10.4 (range: 6 to 14 cycles). These were continued until either progression or unacceptable toxicity occurred, or the patient resigned from chemotherapy.

### 3.2. Response, Progression-Free Survival and Overall Survival

Partial remission (PR) was observed in 11 (55%) patients and 4 (25%) patients showed stable disease (SD). Only four (20%) patients showed progression (PD), including two patients with either an *NRAS* or a *BRAF* mutation (Figure 1).

Median progression-free survival was 9.1 months (95% CI: 7.2–10.1 months; Figure 2). Median overall survival for all patients was 20.7 months (95% CI: 18.2–23.3 months; Figure 3).

### 3.3. Adverse Events

Immediately after TACE, the most common adverse event was abdominal pain, which mostly subsided within a few hours. Nausea and vomiting were less common. In one case, symptoms of cholecystitis were found after the third TACE procedure, which resolved after conservative treatment—this patient continued with FOLFIRI chemotherapy only. After 55 TACE procedures, hematological abnormalities (grades 1–3) were found to have occurred within 7 days after TACE treatment in 19 patients, including with neutropenia or lymphopenia or some with thrombocytopenia, resulting in the postponement of the next cycle of FOLFIRI for 7 days and the next TACE treatment for 18 patients. Delayed diarrhea, treated with loperamide, was found in one patient three days after the second TACE treatment. After consultation with an oncologist and due to concomitant neutropenia in this patient, FOLFIRI chemotherapy without TACE was continued after symptoms resolved. There were no deaths or AEs with CTCAE grade > 3 during or within one month of treatment with TACE (Table 2). 

With FOLFIRI chemotherapy continued alone (after the TACE procedures), grade 4 neutropenia was found in two patients; grade 3 in four patients; and grade 3 thrombocytopenia was found in two patients.

## 4. Discussion

Colorectal cancer (CRC) is one of the most common cancers and a significant body of research, aimed at improving its treatment, has been documented. The prognosis of stage IV CRC is poor, with a 5-year survival rate of 15%. Up to 50% of patients with localized disease will develop metastases [9]. In cases of inoperable metastases from colorectal cancer, a breakthrough that has increased the effectiveness of first- and second-line systemic chemotherapy is the inclusion of monoclonal antibodies (anti-EGRF or anti-VEGF) [10,11]. A *KRAS* (Kirsten rat sarcoma viral oncogene)-activating mutation is a potent mechanism of resistance to EGFR inhibitors and occurs in around 40% of patients with advanced CRC, having a negative impact on overall survival (OS) [12]. *BRAF*-mutated metastatic CRCs (mCRCs) are also associated with poorer survival outcomes and higher rates of distant lymph node metastases [13]. Previously, researchers explored adding targeted therapies to traditional first-line chemotherapies and demonstrated improved outcomes [14,15]. However, the use of almost all of these drugs is contraindicated in patients at a high risk of thromboembolic events, which are frequent complications and give a poor prognosis. The anti-VEGF antibody bevacizumab was one of the first biologic therapies used for treating patients with mCRC regardless of *RAS* mutational status. The use of bevacizumab in combination with chemotherapy is considered a standard first-line treatment for patients with *RAS*-mutant mCRC [16]. However, bevacizumab is contraindicated in patients with severe arterial hypertension or recent artery or venous thromboembolic disease. Patients receiving bevacizumab in combination with chemotherapy with a history of arterial thromboembolism, diabetes or age greater than 65 years have an increased risk of developing arterial complications including myocardial infarctions and cerebrovascular incidents [17,18]. Therefore, it is important to look for alternative ways to increase the effectiveness of treatment in these patients. One possibility might be chemoembolization with microspheres loaded with irinotecan, especially in the first or second line of treatment.

In the present study, the disease control rate was 80%. Patients from our study exhibited both longer median overall survival times and median progression-free survival times compared to patients with *RAS* mutations who received standard FOLFIRI therapy as first-line treatment in the multicenter CRYSTAL trial [19]. However, our values were similar to those in the FIRE-3 trial in patients with *KRAS* mutations, where bevacizumab was combined with first-line FOLFIRI chemotherapy [20]. In our opinion, the positive effects of combining DEB-TACE with FOLFIRI can be attributed to the embolic effect of DEB-TACE, which influences both tumor vasculature degradation and significantly alters the pharmacodynamics of irinotecan within the liver, as compared to FOLFIRI alone. 

The short-term vascular near-stasis achieved during the DEBTACE procedure results in both a more efficient conversion of irinotecan to its metabolite SN-38 (7-Ethyl-10-hydroxycamptothecin), which is several hundred times more potent, and a delay of its washout, prolonging contact time with tumor cells [21,22]. The use of chemoembolization in the first line of treatment also avoids tumor-cell resistance to irinotecan, resulting in greater efficacy.

The main problem with the simultaneous use of irinotecan chemoembolization with FOLFIRI systemic chemotherapy may be its side effects, especially those caused by a cumulatively increased dose of irinotecan [23]. Among side effects found immediately after treatment, there were episodes of abdominal pain requiring analgesic inclusion over several hours, and post-embolization syndrome lasting up to several days. In addition to the typical side effects from chemoembolization such as abdominal pain, asthenia and subfebrile states, hematological disorders (especially leukopenia) may also occur [24]. The presence of significant side effects has been demonstrated in a study that combined irinotecan chemoembolization with mFOLFOX6 chemotherapy [25]. 

The percentage of hematological adverse events (AEs) in our study was higher than that observed with standard FOLFIRI chemotherapy, but these were predominantly grade 1-3 AEs [26]. For most patients, these AEs resulted in the necessity of delaying the next DEB-TACE or subsequent FOLFIRI treatment by one week. However, the average number of chemotherapy cycles per patient was comparable to patients undergoing a standard FOLFIRI regimen [27].

A study that evaluated the tolerability of combining chemoembolization with FOLFIRI showed no significant differences in side effects depending on the use of a full or reduced dose of irinotecan [28]. The microspheres found in present chemoembolization therapies release irinotecan slowly, which translates into lower serum concentrations and fewer side effects. Moreover, in order to avoid permanent occlusion of the hepatic arteries, in some procedures, the full dose of irinotecan-loaded microspheres is not administered. This is especially true for procedures involving the left hepatic lobe, due to its smaller vasculature and volume. 

Considering that, in most cases, the metastatic liver lesion stage determines the length of survival and patient comfort, the use of individualized, liver-targeted chemoembolization may be beneficial in some patients. This seems especially true for patients who do not qualify for molecular treatments, for whom a combination of chemoembolization with FOLFIRI systemic chemotherapy may be an attractive therapeutic option.

## 5. Conclusions

The combination of FOLFIRI systemic chemotherapy with chemoembolization procedures for liver metastatic lesions from colorectal cancer may provide a valuable treatment option for patients who do not qualify for monoclonal antibody therapy.

## 6. Limitations

This was a single-center study that included a small sample of patients. Patients were not differentiated by cause of death. 

## Figures and Tables

**Figure 1 cancers-16-00071-f001:**
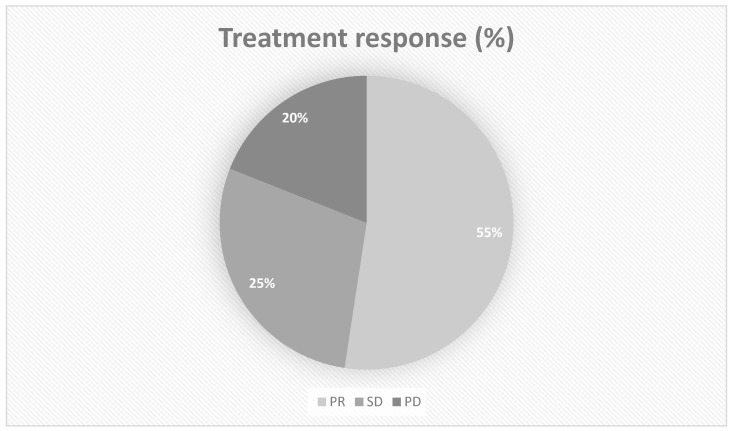
Response to treatment. PR = partial remission; SD = stable disease; PD = progression in disease.

**Figure 2 cancers-16-00071-f002:**
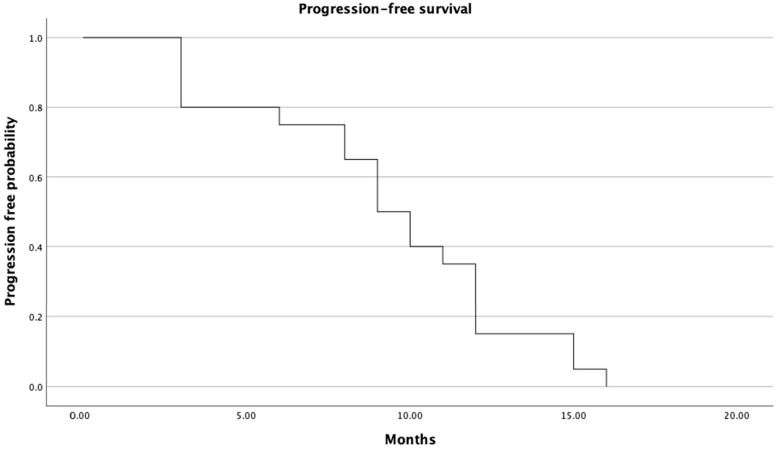
Progression-free survival of all patients.

**Figure 3 cancers-16-00071-f003:**
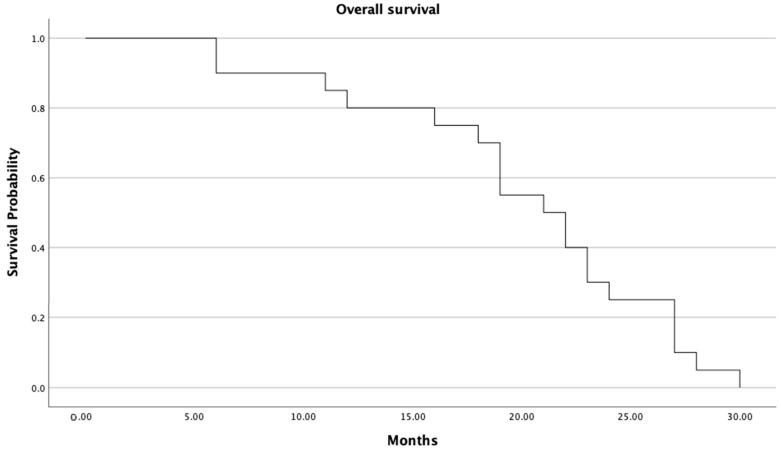
Overall survival of all patients.

**Table 1 cancers-16-00071-t001:** Patient characteristics.

Parameter	Value (*N* = 20)
Age, median (range)	66.2 (37–81)
Sex (*n* (%))	
Female	12 (60%)
Male	8 (40%)
ECOG status (*n* (%))	
0	9 (45%)
1	11 (55%)
Tumor distribution (*n* (%))	
Bilobar	18 (90%)
Unilobar	2 (10%)
Number of liver metastases per patient, median (range)	6.6 (2–9)
Largest nodule size diameter per patient, median (cm)	6.3
Extent of liver involvement (*n*, <25%/>25%)	14/6
Site of primary tumor (*n*)	
Left colon	8
Right colon	12
Molecular status (*n*)	
*KRAS* gene mutated	18
*BRAF* gene mutated	1
*NRAS* gene mutated	1
Carcinoembryonic antigen level (*n*)	
≤15 ng/mL	14
>15 ng/mL	6

*N* = total number of patients; n = number of patients in each category; ECOG = Eastern Cooperative Oncology Group performance status.

**Table 2 cancers-16-00071-t002:** Adverse events after TACE procedures, graded according to the Cancer Therapy Evaluation Program’s Common Terminology Criteria for Adverse Events.

Adverse Event	Number of Patients		
	Grade 1	Grade 2	Grade 3
Abdominal Pain	4	7	4
Nausea	8	7	
Vomiting	5	5	
Fever	4	3	1
Diarrhea	1		1
Neutropenia	1	8	3
Lymphopenia	5	4	1
Thrombocytopenia	8	6	2
Anemia	10	10	
Cholecystitis		1	
Elevated AST	5	1	
Elevated ALT	2		

## Data Availability

The data presented in this study are available on request from the corresponding author.

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
