# Peer review of "Efficacy and Safety of Liver Chemoembolization Procedures, Combined with FOLFIRI Chemotherapy, in First-Line Treatment of Metastatic Colorectal Cancer in Patients with Oncogene Mutations"

_cancers, 2023, doi:10.3390/cancers16010071_

Round 1

Reviewer 1 Report (New Reviewer)

Comments and Suggestions for Authors

The study holds significant importance in the quest for improved options in cancer chemotherapy. Nonetheless, there are several issues with the manuscript that require attention:

Language Quality: The English in the manuscript is weak and should undergo revision by a native English speaker. An example of this is evident in the first sentence of the results section.

Clarity of Ideas and Procedures: The overall concept and methodology of the study lack clarity, making it challenging for readers, especially those without specialized knowledge, to comprehend the content seamlessly.

Absence of Controls: The study lacks proper control groups to enable a meaningful comparison and assessment of the method's effectiveness. This absence of controls is not mitigated by a comparative analysis with other studies utilizing conventional methods.

Discussion Focus: The discussion section appears more akin to a literature review and introduction rather than a focused exploration of the study's results. It is imperative that the discussion concentrates on the outcomes of this specific study.

Methodology: While the authors assert the use of computed tomography scan and Modified Response Evaluation Criteria in Solid Tumors (mRECIST) for assessing treatment response, there is a conspicuous absence of results derived from this method.

Addressing these issues will significantly enhance the manuscript's overall quality and contribute to the efficacy of the study's communication to the scientific community.

Comments on the Quality of English Language

The English in the manuscript is weak and should undergo revision by a native English speaker. An example of this is evident in the first sentence of the results section.

Author Response

Dear Reviewer

Thank you for the critiques and suggestions.

1.Language Quality: The English in the manuscript is weak and should undergo revision by a native English speaker. An example of this is evident in the first sentence of the results section.

The English language of this article has been checked by an Academic Editor who has corrected errors.

2.Clarity of Ideas and Procedures: The overall concept and methodology of the study lack clarity, making it challenging for readers, especially those without specialized knowledge, to comprehend the content seamlessly.

We have improved and simplified the description of the concept and methodology as recommended.

3.Absence of Controls: The study lacks proper control groups to enable a meaningful comparison and assessment of the method's effectiveness. This absence of controls is not mitigated by a comparative analysis with other studies utilizing conventional methods.

In our single-center study, we were unable to recruit an adequate control group. Due to oncological guidelines, patients with oncogenic mutations, without contraindications, were treated with FOLFIRI plus bevacizumab. We compared the results of our study with multicenter, randomized trials involving subgroup of patients with oncogenic mutations. 

4.Discussion Focus: The discussion section appears more akin to a literature review and introduction rather than a focused exploration of the study's results. It is imperative that the discussion concentrates on the outcomes of this specific study.

We have edited the discussions and added a comparison of the results and side effects of our combination therapy with FOLFIRI.

5.Methodology: While the authors assert the use of computed tomography scan and Modified Response Evaluation Criteria in Solid Tumors (mRECIST) for assessing treatment response, there is a conspicuous absence of results derived from this method.

We have added mRECIST criteria and explained why they are preferred in TACE evaluation.

Reviewer 2 Report (New Reviewer)

Comments and Suggestions for Authors

The manuscript aims to evaluate the combined effect of FOLFIRI chemotherapy and liver chemoembolization procedures in treating metastatic colorectal cancer, and they revealed that the liver chemoembolization procedure would potentiate the FOLFIRI chemotherapy, in which there is a 80% disease control rate. The study is somehow interesting and would benefit the metastatic colorectal cancer patients with oncogenes mutations. 

1. there is no side to side comparison for FOLFIRI only and the combined therapy, the authors should include a paragraph in discussion to talk about the how much improvement of the combined therapy vs FOLFIRI chemotherapy alone.

2. the references should be re-edited and using Arabic numeral instead of Roman numeral.

Author Response

        Dear Reviewer

       Thank you for the critiques and suggestions.

  1. There is no side to side comparison for FOLFIRI only and the combined therapy, the authors should include a paragraph in discussion to talk about the how much improvement of the combined therapy vs FOLFIRI chemotherapy alone.

     In the discussion, we compared the results of our study with multicenter,     randomized trials involving subgroup of patients with oncogenic mutations.

  1. The references should be re-edited and using Arabic numeral instead of Roman numeral.

       Done.

Round 2

Reviewer 1 Report (New Reviewer)

Comments and Suggestions for Authors

The authors have addressed the requested revisions. However, additional improvements are needed to enhance the English proficiency of the manuscript. 

Comments on the Quality of English Language

Needs improvement 

This manuscript is a resubmission of an earlier submission. The following is a list of the peer review reports and author responses from that submission.

Round 1

Reviewer 1 Report

Comments and Suggestions for Authors

Efficacy and safety of liver chemoembolization procedures, combined with FOLFIRI chemotherapy, in first-line treatment of metastatic colorectal câncer

The article seems interesting, but I missed more refined data and analysis. The article brought more qualitative results and a more superficial profile of the participants. In fact, I was unable to identify a difference in the data collected and something that could show an important impact on the disease. Imaging exams were not present and I found the data very superficial.

The article contains 17 references and none from the year 2023. For such an important subject, the references are not very current. The formatting of the article is also flawed. I believe that the template used is outdated, as it is dated 2022.

Comments on the Quality of English Language

English needs technical adjustments.

Reviewer 2 Report

Comments and Suggestions for Authors

I think its contribution to the literature will be limited. It can be evaluated in other journals.